# Volumetric subfield analysis of cynomolgus monkey's choroid derived from hybrid machine learning optical coherence tomography segmentation

**Peter M. Maloca**[1,2,3]*, **Philippe Valmaggia**[1,2], **Theresa Hartmann**[4], **Marlene Juedes**[4], **Pascal W. Hasler**[2], **Hendrik P. N. Scholl**[1,2], **Nora Denk**[1,4]

**1** Institute of Molecular and Clinical Ophthalmology Basel (IOB), Basel, Switzerland, **2** Department of Ophthalmology, University Hospital Basel, Basel, Switzerland, **3** Moorfields Eye Hospital NHS Foundation Trust, London, United Kingdom, **4** Pharma Research and Early Development (pRED), Pharmaceutical Sciences (PS), Roche, Innovation Center Basel, Basel, Switzerland

* peter.maloca@iob.ch

**Data Availability Statement:** All data are made available in the Supporting Information.

## Abstract

This study aimed to provide volumetric choroidal readings regarding sex, origin, and eye side from healthy cynomolgus monkey eyes as a reference database using optical coherence tomography (OCT) imaging. A machine learning (ML) algorithm was used to extract the choroid from the volumetric OCT data. Classical computer vision methods were then applied to automatically identify the deepest location in the foveolar depression. The choroidal thickness was determined from this reference point. A total of 374 eyes of 203 cynomolgus macaques from Asian and Mauritius origin were included in the analysis. The overall subfoveolar mean choroidal volume in zone 1, in the region of the central bouquet, was 0.156 mm$^3$ (range, 0.131–0.193 mm$^3$). For the central choroid volume, the coefficient of variation (CV) was found of 6.3%, indicating relatively little variation. Our results show, based on analyses of variance, that monkey origin (Asian or Mauritius) does not influence choroid volumes. Sex had a significant influence on choroidal volumes in the superior-inferior axis (p $\leq$ 0.01), but not in the fovea centralis. A homogeneous foveolar choroidal architecture was also observed.

## Introduction

The cynomolgus macaque, *Macaca fascicularis*, was introduced to the island of Mauritius several hundred years ago and has since evolved in relative isolation compared to its conspecifics in Asia [1]. Owing to its genetic and morphological similarity of the eye to humans [2, 3]—especially the presence of a fovea, which can be depicted using optical coherence tomography—the cynomolgus macaque has evolved as a commonly used preclinical species in ocular drug development [4–6]. Additionally, in an animal model of achromatopsia, important findings were obtained regarding the tolerability of intraocular injections of recombinant adeno-

**Funding:** This study was supported by Hoffmann–La Roche Ltd. (Basel, Switzerland). The funder provided support in the form of salaries for TH, MJ, and ND, as well as PMM and PH (consultants). The finder did not play any role in the study design, data collection and analysis, decision to publish, or preparation of the manuscript.

**Competing interests:** The funder provided support in the form of salaries for TH, MJ, and ND, as well as PMM and PH (consultants). This does not alter the authors' adherence to the PLOS ONE policies on sharing data and materials. Dr. Hendrik Scholl is supported by the Swiss National Science Foundation (Project funding: "Developing novel outcomes for clinical trials in Stargardt disease using structure/function relationship and deep learning" #310030_201165, and National Center of Competence in Research Molecular Systems Engineering: "NCCR MSE: Molecular Systems Engineering (phase II)" #51NF40-182895), the Wellcome Trust (PINNACLE study), and the Foundation Fighting Blindness Clinical Research Institute (ProgStar study). Dr. Scholl is member of the Scientific Advisory Board of: Astellas Pharma Global Development, Inc./Astellas Institute for Regenerative Medicine; Boehringer Ingelheim Pharma GmbH & Co; Gyroscope Therapeutics Ltd.; Janssen Research & Development, LLC (Johnson & Johnson); Novartis Pharma AG (CORE); Okuvision GmbH; and Third Rock Ventures, LLC. Dr. Scholl is a consultant of: Gerson Lehrman Group; Guidepoint Global, LLC; and Tenpoint Therapeutics Limited. Dr. Scholl is member of the Data Monitoring and Safety Board/Committee of Belite Bio (CT2019-CTN-04690-1), ReNeuron Group Plc/Ora Inc. (NCT02464436), F. Hoffmann-La Roche Ltd (VELODROME trial, NCT04657289; DIAGRID trial, NCT05126966) and member of the Steering Committee of Novo Nordisk (FOCUS trial; NCT03811561). All arrangements have been reviewed and approved by the University of Basel (Universitätsspital Basel, USB) and the Board of Directors of the Institute of Molecular and Clinical Ophthalmology Basel (IOB) in accordance with their conflict of interest policies. Compensation is being negotiated and administered as grants by USB which receives them on its proper accounts. Dr. Scholl is co-director of the Institute of Molecular and Clinical Ophthalmology Basel (IOB) which is constituted as a non-profit foundation and receives funding from the University of Basel, the University Hospital Basel, Novartis, and the government of Basel-Stadt.

associated virus [6]. Other studies have documented breakthroughs in anti-vascular endothelial growth factor therapy in models of neovascular age-related macular degeneration [4, 7].

These studies share a common interest in the morphological changes in the retina. Much emphasis has been placed on the study of the fovea, which is the site of best visual acuity [8, 9]. Interestingly, the fovea of humans and macaques also reveal similar foveal vascular anatomy. Most notably, a central foveolar avascular zone (FAZ) was found in both cases [10, 11]. This is particularly remarkable, because the photoreceptor cells are dependent on a healthy choriocapillaris [12–14].

It can hence be assumed that the fovea is the preferred site for neurodegenerative and circulatory diseases. In part, this particular vulnerability [15] may be caused by its vascular deprivations as the fovea is almost entirely dependent on appropriate blood perfusion across the choroid [16, 17]. Optical coherence tomography (OCT) has made it possible to visualize deep-seated choroidal vessels non-invasively [16, 18, 19].

For example, it has been shown that a single occlusion of a vortex vein is relatively well tolerated, but when two vortex veins are occluded, significant hemodynamic and structural changes occur in the choroid [5]. The dynamics of choroidal changes in the context of medical treatment could also be demonstrated by systemic adrenaline injection increasing, whereas photodynamic therapy with verteporfin reduced subfoveal choroidal thickness [20].

In view of these interesting results, normative cynomolgus macaques' values of choroidal thickness have also been suggested [21]. Overall, these values are very useful, but they were collected from a relatively small sample of cynomolgus macaques and did not compare their origins. This is even more important as significant differences between cynomolgus macaques of different origins have been described for the retina, indicating they are not freely interchangeable for retinal research purposes [22].

Therefore, this study was conducted on an unprecedented number of cynomolgus macaques to determine the influence of sex, origin, and eye site on the volumetric parameters of the choroid for the first time. This allows the measured values to be considered in relation to naturally occurring variations.

## Materials and methods

### Animals and husbandry

This retrospective study analyzed OCT data from studies conducted as part of regular support for routine pharmaceutical product development [22–24]. The aim of these primary investigations was to obtain OCT data for safety assessment, and so the animals were observed consecutively. Only the predose OCT image data from treatment-naïve cynomolgus monkeys of both sexes were retrospectively reviewed for the purpose and use for the current study. No additional animal experimentation was carried out for this study. The primary experiments were reviewed and approved by the Institutional Animal Care and Use Committee (IACUC) of their respective institutions: Charles River Laboratories Montreal, ULC Institutional Animal Care and Use Committee (CR-MTL IACUC), IACUC Charles River Laboratories Reno (OLAW Assurance No. D16-00594), and the IACUC (Covance Laboratories Inc., Madison, WI, USA; OLAW Assurance #D16-00137 A3218-01). The animals were treated and utilized strictly in accordance with the guidelines of either the US National Research Council or the Canadian Council on Animal Care. Animals were group housed in stainless steel cages, according to European housing standards described in Annex III of Directive 2010/63/EU. The animals were bred for use in the laboratory and were made available by certified suppliers from two geographical regions: Mauritius and Asia. Room temperature was maintained constantly between 20˚C and 26˚C, humidity was between 20% and 70%, and the light-dark cycle

was a standard 12:12 h cycle. The animals were fed a standard diet of pellets supplemented with fresh fruits and vegetables. Tap water was offered freely via an automated watering system after being treated by reverse osmosis and ultraviolet irradiation. Psychological and environmental enrichment was provided to animals except during study procedures and activities.

All study protocols and any amendments or procedures complied with the ARVO Statement for the Use of Animals in Ophthalmic and Vision Research, with all studies reviewed and approved in advance by the institutional animal welfare and use committees of the respective institutions.

## Animals

Thus, data were available from healthy, untreated cynomolgus monkeys of Mauritian or Asian origin. Their ages ranged from 30 to 50 months, and they weighed between 2.5 and 5.5 kg.

## OCT image acquisition

Spectral OCT data were collected using a dilated pupil (Heidelberg Engineering, Heidelberg, Germany) [22]. The scanning protocol consisted of horizontal line scans (scan size 20˚) and 25 raster lines (spacing 221 μm, scan length 5.3 mm, 512 × 496 pixels, and scan depth 1.9 mm). All OCT data were exported from the unit in bitmap image data format (BMP).

## Machine learning and image processing

Image processing has been described in detail earlier [22]. The machine learning (ML) algorithm used in this work for semantic image segmentation has also been reported previously along with a description of its accuracy [25]. In short, the machine learning algorithm is a scalable, deep learning-based algorithm which creates semantic image segmentations of B-scans [25]. For every pixel of a B-scan it predicts the eye compartment, i.e., vitreous, retina, choroid, or sclera (Fig 1A). This study used the deep learning-based algorithm to identify the choroid compartment. The deep learning-based algorithm generated a semantic image segmentation of a B-scan within seconds, whereas a human grader generally needs longer and gets tired after some time.

In the second step, a classical computer vision algorithm was used to automatically determine and define the deepest location of the foveolar depression within the OCT volume, which was designated as the nulla [22]. All further measurements were performed based on this reference point. The nulla is of particular importance for foveolar depression because at this site, the most direct interaction of the photoreceptors with light is possible. Based on the nulla, a rectangular region of interest (ROI) was placed in the B-scan plane, with a total width of 3000 μm (Fig 1). The ROI defined the longitudinal section of a cylindrical region (Fig 1), which was used to determine the volumetric regions. This study determined the volumes of three concentric zones (Z1 –Z3), four quadrants (Q1 –Q4), and nine slices (S1 –S9), as shown in Fig 1. Zones 1 and slice 1 represent the same volume. Thus, only zone 1 was included in the results, and slice 1 was omitted.

## Statistical analysis

For each of the measured volumes, the summary statistics mean, median, standard deviation, minimum, maximum, and coefficient of variation were calculated for subgroups of the data (e.g., for females of Asian origin). Additionally, for zone 1 (the region of the central bouquet of cones), the overall summary statistics were calculated based on all eyes.

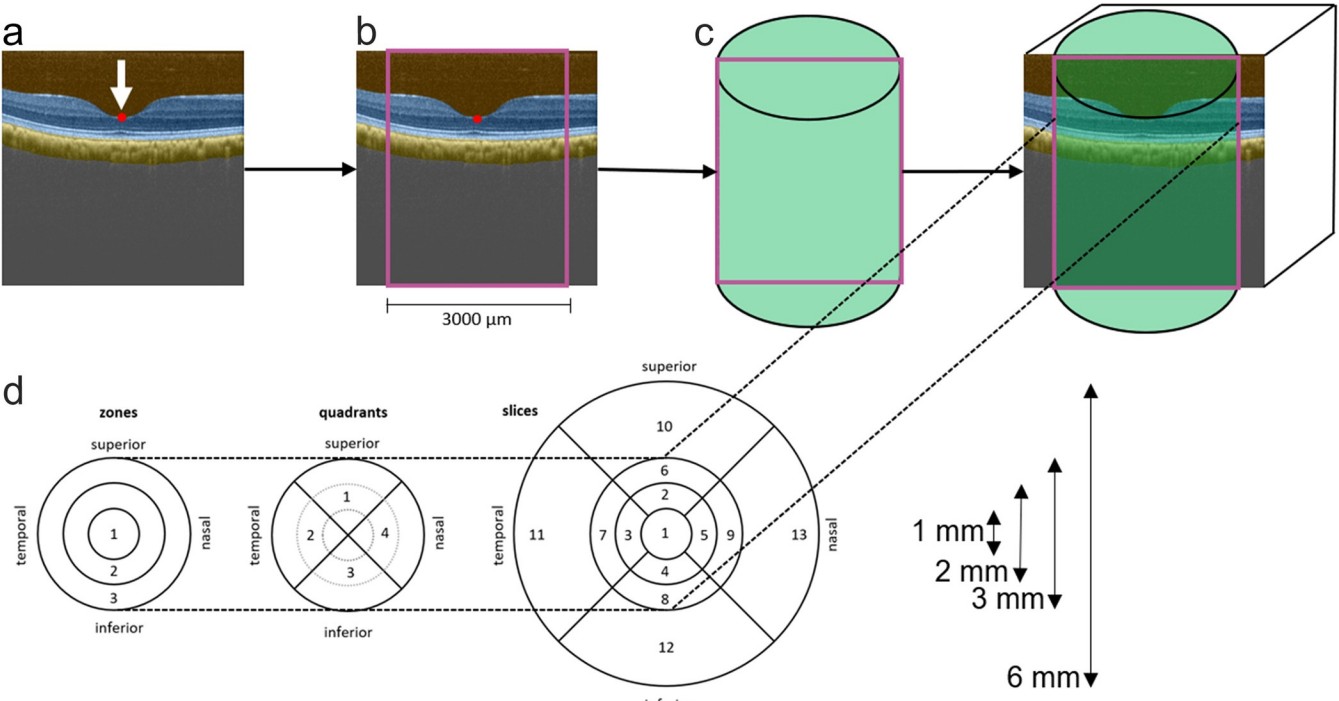

**Fig 1. Choroidal volume measurements in a right eye. a.** A machine learning algorithm was trained to detect the choroid from an obtained macula volume OCT scan (highlighted in yellow; brown = vitreous, blue = retina). For a better overview, only a single B-scan is illustrated here. Consequently, a classic algorithm automatically defined the deepest location within the foveolar depression which was marked a nulla (arrow, red spot). **b.** Starting from nulla, a rectangle (depicted in pink) was defined to the side with a total length of 3000 μm. **c.** This rectangle was rotated axially centered on nulla to segment the choroid within the OCT volume allowing measurements only the central and paracentral subfields. **d.** From the segmented choroid volume, choroidal sub-fields were analyzed, marked as circular zones, quadrants, and slices. (outer zones 10–13 were not investigated).

Pearson correlation analysis was performed on all eyes to investigate the correlation between Z1 and Z3, Q1–Q4, and S1–S9. Principal Component Analysis (PCA) was performed to investigate the patterns of variability in the data for S1–S9. The aim of PCA was to identify latent "factors," which can explain the variability in the data. Pearson correlation analysis and PCA were performed with Python libraries pandas v1.2.0 and statsmodels v0.12.1, respectively.

A Multivariate Analyses of Variance (MANOVA) was conducted to test the effect of the independent variables "sex," "origin," and their interaction on the nine dependent variables S1—S9 jointly. This analysis intended to detect "overall" effects of "sex" and "origin." For the MANOVA analysis, there should not be outliers in the independent variables. For the right and left eyes, one and eight outliers, respectively, were excluded from the MANOVA analyses. Outliers were detected by quantile-quantile plots that plotted observed Mahalanobis distances of data points to the multidimensional mean against expected Mahalanobis distances to the multidimensional mean, which are supposed to follow a $\chi^2$ distribution. Sixteen eyes of unknown origin were excluded from MANOVA analyses. p-values were calculated using the F statistic, which is part of statsmodels' MANOVA implementation. Significance level 0.01 was used to identify significant effects.

An individual Analysis of Variance (ANOVA) investigated the effect of the independent variables "sex" and "origin" on each of the nine dependent variables S1—S9 individually. Sixteen eyes of unknown origin were excluded from the ANOVA analyses. ANOVA and MANOVA were performed using Python library statsmodels v0.12.1. For ANOVA, the significance of differences between group means was generated using the F statistic, which is part of

statsmodels' ANOVA implementation. The Bonferroni correction was performed by dividing the significance levels by nine (the number of individual ANOVA analyses) to counteract the multiple testing problem. The significance levels 0.05/9, 0.01/9, and 0.001/9 were used to report the significance of effects.

Boxplots were used to visualize the distribution of the data and for group-wise comparisons (e.g., Mauritius versus Asian origin). Boxplots were created using the Python library seaborn v0.11.1. All statistical analyses and visualizations were done in Python v3.8.5.

## Results

### General results

In total, volumetric OCT data were collected from 374 eyes originating from 203 different cynomolgus monkeys. Females contributed 147 eyes (39.30%), and males contributed 227 eyes (60.70%). 186 eyes were left eyes (49.73%), and 188 eyes were right eyes (50.27%). Monkeys of Mauritius origin contributed 199 eyes (53.20%), and monkeys of Asian origin contributed 159 eyes (46.80%). Sixteen eyes of male individuals were of an unknown origin.

### Summary statistics

For zone 1, the region of the sharpest vision, an overall analysis including all 374 eyes revealed a mean volume of 0.156 mm$^3$ (range, 0.131 to 0.193 with a CV of 6.3%). A subgroup analysis dividing the animals according to sex and origin revealed generally similar distributions of the measured coefficients (Figs 1–3 and Tables 1–3). However, some systematic differences with respect to sex were observed (e.g., in Z3, Q3, or S8). The results are summarized in Figs 2–4.

### Correlation analysis

Pearson correlation analysis (Table 4) revealed that the mean correlation among zones was 0.81 (0.76–0.87), among quadrants 0.84 (0.82–0.88), and among the slices 0.67 (0.17–0.91). Furthermore, zones and quadrants showed a mean correlation to slices of 0.78 (0.48–1.00) and 0.78 (0.52–0.98), respectively. The zone and quadrant coefficients were mostly composed of nine slice coefficients (Fig 1). Thus, to keep the number of statistical hypothesis tests small and to counteract the multiple testing problem, only nine slice coefficients S1–S9 were used in further statistical analyses.

### Principal components analysis

The PCA yielded largely similar results in right and left eyes (Fig 5A and 5C, Table 5). The first two principal components (PCs) explained 88.1% and 88.8% of the variability in right and left eye, respectively (Fig 5A and 5C).

The first PC is the average of the nine slices, with the absolute values of slice coefficients on the nasal-temporal axis being slightly larger than those on the superior-inferior axis (Table 5). The second PC was a center-vs-edge factor on the superior-inferior axis (Table 5). It assigns relatively large weights to slices at the edges (S6, S8) and relatively large negative weights to slices near the center (S2, S4). However, the center slide itself (S1) receives weights near zero.

### Statistical hypothesis tests

**MANOVA analysis.** MANOVA was performed to investigate the effects of sex, origin, and their interaction on the nine slice coefficients (S1 –S9) (Table 6). A MANOVA was performed for the right (first two rows) and left (bottom two rows) eyes separately. In contrast to origin, sex has a significant effect on the dependent variables S1–S9 in right and left eyes at

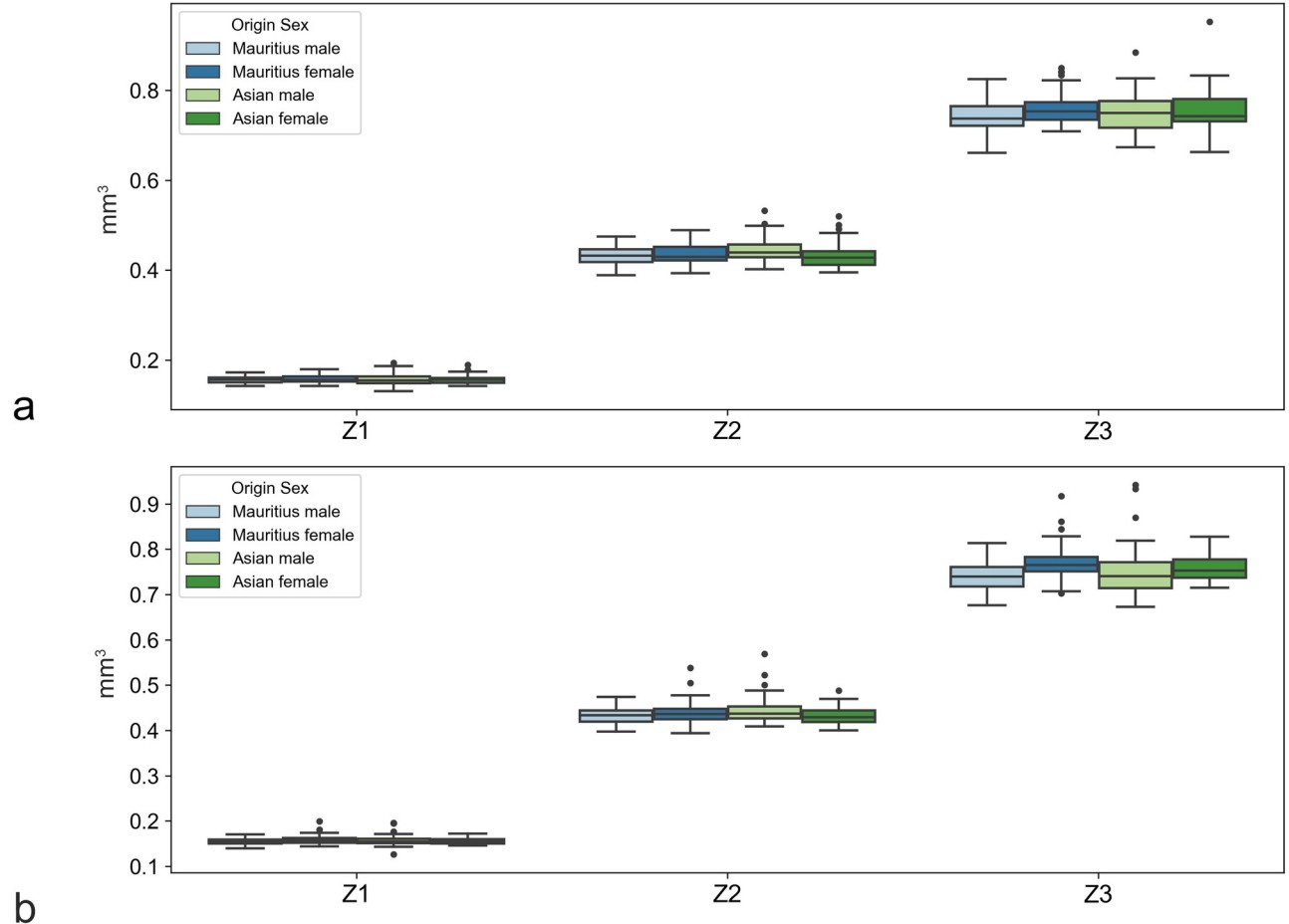

**Fig 2. Boxplots of sex-specific variations in choroid volumes.** For right (**2a**, OD) and left eyes (**2b**, OS) measured as circular zones centered on the foveolar depression.

significance level 0.01. Effect size is measured using Wilks' lambda. The test results are equivalent with Pillai's trace, Hotelling–Lawley trace, and Roy's greatest root. A joint analysis of the nine slice coefficients (S1 –S9) appears reasonable because the nine slice coefficients are correlated with each other (see correlation analysis). MANOVA was performed separately for the right and left eyes. Interaction terms were not significant at a significance level of 0.01, and thus were removed from the models. Sex had a significant effect on both eyes whereas origin did not have a significant effect (significance level 0.01).

**ANOVA analysis.** The results of the statistical significance tests based on ANOVA with regard to sex and origin are summarized in Table 7. The interaction terms were not significant at a significance level of 0.01 / 9 and were thus removed from the models. The origin did not have a significant influence on slice volumes (significance level 0.05 / 9). In contrast, sex showed a significant influence along the superior-inferior axis on slice volumes S2, S4, S6, and S8 (significance level 0.05 / 9). The results largely agreed between the right and left eye.

## Discussion

Numerous diseases can be associated with changes in the choroid, such as age-related macular degeneration, glaucoma, and diabetic retinopathy, which can also be examined using OCT

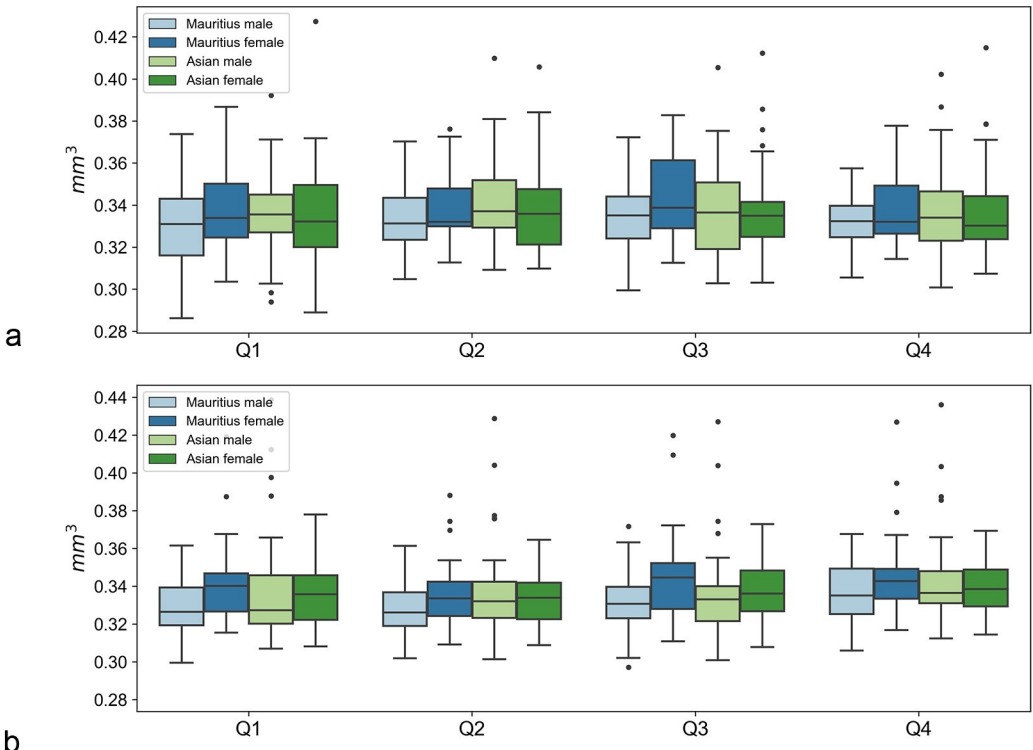

**Fig 3. Boxplots of sex-specific variations in choroid volumes.** For right (**3a**, OD) and left eyes (**3b**, OS) measured as quadrants.

**Table 1. Summary presentation of the choroid volume zone values male compared to female monkeys and origin.**

| | | Zone 1 | | | | | Zone 2 | | | | | Zone 3 | | | | |
|---|---|---|---|---|---|---|---|---|---|---|---|---|---|---|---|---|
| | **Stats** | **all** | **m/M** | **m/A** | **f/M** | **f/A** | **all** | **m/M** | **m/A** | **f/M** | **f/A** | **all** | **m/M** | **m/A** | **f/M** | **f/A** |
| **OD** | count | 188 | 62 | 45 | 34 | 39 | 188 | 62 | 45 | 34 | 39 | 188 | 62 | 45 | 34 | 39 |
| | mean | 0.156 | 0.156 | 0.156 | 0.157 | 0.156 | 0.436 | 0.433 | 0.445 | 0.436 | 0.432 | 0.751 | 0.741 | 0.750 | 0.762 | 0.759 |
| | std | 0.010 | 0.008 | 0.014 | 0.009 | 0.010 | 0.025 | 0.021 | 0.027 | 0.024 | 0.029 | 0.042 | 0.036 | 0.044 | 0.040 | 0.050 |
| | min | 0.131 | 0.142 | 0.131 | 0.143 | 0.142 | 0.389 | 0.389 | 0.402 | 0.393 | 0.395 | 0.661 | 0.661 | 0.673 | 0.708 | 0.663 |
| | median | 0.156 | 0.156 | 0.155 | 0.155 | 0.157 | 0.432 | 0.432 | 0.439 | 0.430 | 0.428 | 0.746 | 0.737 | 0.750 | 0.753 | 0.742 |
| | max | 0.193 | 0.172 | 0.193 | 0.179 | 0.189 | 0.532 | 0.475 | 0.532 | 0.489 | 0.520 | 0.952 | 0.825 | 0.885 | 0.850 | 0.952 |
| | CV | 0.063 | 0.049 | 0.086 | 0.056 | 0.061 | 0.057 | 0.047 | 0.059 | 0.054 | 0.066 | 0.056 | 0.048 | 0.057 | 0.051 | 0.065 |
| **OS** | count | 186 | 65 | 39 | 38 | 36 | 186 | 65 | 39 | 38 | 36 | 186 | 65 | 39 | 38 | 36 |
| | mean | 0.156 | 0.155 | 0.157 | 0.158 | 0.156 | 0.435 | 0.433 | 0.445 | 0.439 | 0.432 | 0.750 | 0.739 | 0.753 | 0.770 | 0.758 |
| | std | 0.010 | 0.007 | 0.014 | 0.010 | 0.007 | 0.025 | 0.020 | 0.031 | 0.027 | 0.020 | 0.044 | 0.033 | 0.058 | 0.044 | 0.029 |
| | min | 0.126 | 0.139 | 0.126 | 0.144 | 0.146 | 0.382 | 0.397 | 0.409 | 0.394 | 0.400 | 0.643 | 0.677 | 0.673 | 0.702 | 0.714 |
| | median | 0.155 | 0.154 | 0.155 | 0.157 | 0.155 | 0.434 | 0.434 | 0.437 | 0.436 | 0.429 | 0.748 | 0.740 | 0.740 | 0.765 | 0.753 |
| | max | 0.199 | 0.170 | 0.196 | 0.199 | 0.172 | 0.569 | 0.474 | 0.569 | 0.538 | 0.488 | 0.942 | 0.814 | 0.942 | 0.917 | 0.827 |
| | CV | 0.063 | 0.045 | 0.087 | 0.064 | 0.043 | 0.058 | 0.045 | 0.070 | 0.060 | 0.046 | 0.058 | 0.044 | 0.076 | 0.057 | 0.037 |

OD = oculus dexter, OS = oculus sinister, Stats = statistic, std = standard deviation, min = minimum, max = maximum, CV = coefficient of variation, m = male, f = female, M = Mauritius, A = Asian, values in mm$^3$. Note that slice 1 is identical to zone 1.

**Table 2. Summary presentation of the choroid volume quadrant values male compared to female monkeys and origin.**

| | Stats | Quadrant 1 | | | | | Quadrant 2 | | | | | Quadrant 3 | | | | | Quadrant 4 | | | | |
|---|---|---|---|---|---|---|---|---|---|---|---|---|---|---|---|---|---|---|---|---|---|
| | | all | m/M | m/A | f/M | f/A | all | m/M | m/A | f/M | f/A | all | m/M | m/A | f/M | f/A | all | m/M | m/A | f/M | f/A |
| OD | count | 188 | 62 | 45 | 34 | 39 | 188 | 62 | 45 | 34 | 39 | 188 | 62 | 45 | 34 | 39 | 188 | 62 | 45 | 34 | 39 |
| | mean | 0.334 | 0.330 | 0.336 | 0.337 | 0.337 | 0.337 | 0.333 | 0.342 | 0.337 | 0.338 | 0.337 | 0.334 | 0.337 | 0.343 | 0.337 | 0.335 | 0.333 | 0.337 | 0.337 | 0.336 |
| | std | 0.021 | 0.019 | 0.020 | 0.021 | 0.024 | 0.019 | 0.015 | 0.021 | 0.016 | 0.022 | 0.020 | 0.016 | 0.021 | 0.020 | 0.022 | 0.018 | 0.013 | 0.020 | 0.017 | 0.021 |
| | min | 0.286 | 0.286 | 0.294 | 0.304 | 0.289 | 0.305 | 0.305 | 0.309 | 0.313 | 0.310 | 0.299 | 0.299 | 0.303 | 0.313 | 0.303 | 0.301 | 0.306 | 0.301 | 0.314 | 0.307 |
| | median | 0.333 | 0.331 | 0.336 | 0.334 | 0.332 | 0.334 | 0.331 | 0.337 | 0.332 | 0.336 | 0.336 | 0.335 | 0.336 | 0.339 | 0.335 | 0.332 | 0.332 | 0.334 | 0.332 | 0.330 |
| | max | 0.427 | 0.374 | 0.392 | 0.387 | 0.427 | 0.410 | 0.370 | 0.410 | 0.376 | 0.406 | 0.412 | 0.372 | 0.405 | 0.383 | 0.412 | 0.415 | 0.358 | 0.402 | 0.378 | 0.415 |
| | CV | 0.062 | 0.057 | 0.058 | 0.061 | 0.070 | 0.055 | 0.045 | 0.061 | 0.048 | 0.063 | 0.058 | 0.047 | 0.062 | 0.058 | 0.065 | 0.053 | 0.040 | 0.060 | 0.051 | 0.063 |
| OS | count | 186 | 65 | 39 | 38 | 36 | 186 | 65 | 39 | 38 | 36 | 186 | 65 | 39 | 38 | 36 | 186 | 65 | 39 | 38 | 36 |
| | mean | 0.333 | 0.329 | 0.337 | 0.341 | 0.335 | 0.332 | 0.329 | 0.338 | 0.335 | 0.333 | 0.336 | 0.332 | 0.336 | 0.344 | 0.338 | 0.340 | 0.336 | 0.343 | 0.346 | 0.340 |
| | std | 0.021 | 0.015 | 0.029 | 0.021 | 0.016 | 0.018 | 0.014 | 0.025 | 0.018 | 0.013 | 0.020 | 0.015 | 0.024 | 0.023 | 0.015 | 0.019 | 0.015 | 0.025 | 0.021 | 0.013 |
| | min | 0.288 | 0.299 | 0.307 | 0.316 | 0.308 | 0.288 | 0.302 | 0.301 | 0.309 | 0.309 | 0.282 | 0.297 | 0.301 | 0.311 | 0.308 | 0.306 | 0.306 | 0.312 | 0.317 | 0.314 |
| | median | 0.329 | 0.327 | 0.327 | 0.340 | 0.336 | 0.331 | 0.326 | 0.332 | 0.334 | 0.334 | 0.334 | 0.331 | 0.333 | 0.345 | 0.336 | 0.337 | 0.335 | 0.336 | 0.343 | 0.339 |
| | max | 0.439 | 0.362 | 0.439 | 0.419 | 0.378 | 0.429 | 0.361 | 0.429 | 0.388 | 0.365 | 0.427 | 0.372 | 0.427 | 0.420 | 0.373 | 0.436 | 0.368 | 0.436 | 0.427 | 0.369 |
| | CV | 0.063 | 0.045 | 0.084 | 0.059 | 0.047 | 0.055 | 0.041 | 0.072 | 0.052 | 0.039 | 0.060 | 0.045 | 0.071 | 0.066 | 0.044 | 0.056 | 0.045 | 0.071 | 0.059 | 0.039 |

OD = oculus dexter, OS = oculus sinister, Stats = statistics, std = standard deviation, min = minimum, max = maximum, CV = coefficient of variation, m = male

f = female, M = Mauritius, A = Asian, values in mm$^3$. Note that slice 1 is identical to zone 1.

[26–31]. In this context, cynomolgus monkeys show particular advantages as animal models, as they have a similar structure to the human eye, including the presence of a fovea and choroid [2, 3]. This is important because the fovea, as the site of sharpest vision, is highly dependent on a healthy choroid [32].

Therefore, this study aimed to investigate, for the first time, the natural variation in choroidal volume in healthy cynomolgus monkey eyes by using automated methods in an unprecedented number of eyes. Interestingly, it was revealed that origin did not influence choroid volumes. Neither the MANOVA nor ANOVA analyses revealed a significant effect of origin on any of the slice volumes (S1 –S9). This could indicate that, at the population level, choroidal volume is a relatively fixed trait since it does not differ between the Mauritius and Asian populations. In contrast, sex influences choroid volumes. In other words, there is sexual size dimorphism in cynomolgus monkeys with respect to choroid volumes. ANOVA analyses showed that sex had a significant effect on the slices that were located along the superior-inferior axis, except for the most central slice S1. Interestingly, sex did not have a significant effect on the slices located on the temporal-nasal axis.

Additionally, the coefficient of variation (CV) differed between slices located on the superior-inferior and slices located on the temporal-nasal axes. The CV score of the slices on the superior-inferior axis was between 0.71 and 0.78. The CV score of the slices on the temporal-nasal axis was between 0.52 and 0.58. Thus, besides the significant effect of sex on slices on the superior-inferior axis, there was also more variation along the superior-inferior axis than along the temporal-nasal axis. The CV score of the most central area, Z1 or S1, was 0.063 (in both eyes).

The central area of the choroid, Z1 or S1—which is closest to the foveolar cones—was influenced by neither origin nor sex. This is in complete contrast to observations of retinal structure in the same study population [22]. Thus, the central choroid shows a structural blueprint that is maintained across sexes and origins to provide the fovea with nutrients and adequate metabolites. This suggests that OCT recordings of the central choroid (Z1) can be used independently of the origin and sex in preclinical animal studies.

**Table 3. Summary presentation of the choroid volume slice values male compared to female monkeys and origin.**

| | | Slice 2 | | | | | Slice 3 | | | | | Slice 4 | | | | | Slice 5 | | | | |
|---|---|---|---|---|---|---|---|---|---|---|---|---|---|---|---|---|---|---|---|---|---|
| | Stats | all | m/M | m/A | f/M | f/A | all | m/M | m/A | f/M | f/A | all | m/M | m/A | f/M | f/A | all | m/M | m/A | f/M | f/A |
| OD | count | 188 | 62 | 45 | 34 | 39 | 188 | 62 | 45 | 34 | 39 | 188 | 62 | 45 | 34 | 39 | 188 | 62 | 45 | 34 | 39 |
| | mean | 0.106 | 0.105 | 0.110 | 0.104 | 0.104 | 0.112 | 0.111 | 0.112 | 0.113 | 0.113 | 0.107 | 0.107 | 0.111 | 0.106 | 0.104 | 0.112 | 0.110 | 0.112 | 0.113 | 0.112 |
| | std | 0.008 | 0.008 | 0.009 | 0.007 | 0.008 | 0.006 | 0.005 | 0.007 | 0.006 | 0.008 | 0.008 | 0.006 | 0.008 | 0.007 | 0.008 | 0.006 | 0.005 | 0.007 | 0.006 | 0.007 |
| | min | 0.090 | 0.091 | 0.094 | 0.090 | 0.093 | 0.099 | 0.100 | 0.099 | 0.104 | 0.102 | 0.090 | 0.095 | 0.099 | 0.094 | 0.090 | 0.097 | 0.100 | 0.097 | 0.105 | 0.102 |
| | median | 0.105 | 0.104 | 0.108 | 0.104 | 0.103 | 0.111 | 0.110 | 0.112 | 0.112 | 0.112 | 0.105 | 0.105 | 0.109 | 0.105 | 0.103 | 0.111 | 0.110 | 0.111 | 0.111 | 0.111 |
| | max | 0.131 | 0.122 | 0.131 | 0.119 | 0.127 | 0.135 | 0.125 | 0.133 | 0.127 | 0.135 | 0.136 | 0.122 | 0.136 | 0.123 | 0.124 | 0.139 | 0.120 | 0.132 | 0.127 | 0.139 |
| | CV | 0.078 | 0.071 | 0.078 | 0.068 | 0.077 | 0.058 | 0.047 | 0.066 | 0.051 | 0.066 | 0.071 | 0.059 | 0.071 | 0.067 | 0.078 | 0.056 | 0.045 | 0.063 | 0.053 | 0.066 |
| OS | count | 186 | 65 | 39 | 38 | 36 | 186 | 65 | 39 | 38 | 36 | 186 | 65 | 39 | 38 | 36 | 186 | 65 | 39 | 38 | 36 |
| | mean | 0.105 | 0.106 | 0.110 | 0.105 | 0.103 | 0.111 | 0.110 | 0.112 | 0.113 | 0.112 | 0.106 | 0.107 | 0.110 | 0.106 | 0.104 | 0.112 | 0.111 | 0.113 | 0.115 | 0.113 |
| | std | 0.008 | 0.007 | 0.009 | 0.007 | 0.007 | 0.006 | 0.005 | 0.009 | 0.006 | 0.005 | 0.008 | 0.006 | 0.009 | 0.008 | 0.007 | 0.007 | 0.005 | 0.009 | 0.007 | 0.004 |
| | min | 0.088 | 0.092 | 0.097 | 0.093 | 0.093 | 0.096 | 0.101 | 0.099 | 0.103 | 0.103 | 0.090 | 0.095 | 0.098 | 0.090 | 0.090 | 0.099 | 0.102 | 0.099 | 0.104 | 0.105 |
| | median | 0.106 | 0.106 | 0.109 | 0.105 | 0.103 | 0.110 | 0.109 | 0.110 | 0.112 | 0.112 | 0.106 | 0.107 | 0.108 | 0.106 | 0.103 | 0.112 | 0.111 | 0.111 | 0.114 | 0.113 |
| | max | 0.143 | 0.120 | 0.143 | 0.127 | 0.128 | 0.141 | 0.121 | 0.141 | 0.132 | 0.122 | 0.146 | 0.122 | 0.146 | 0.136 | 0.117 | 0.143 | 0.121 | 0.139 | 0.143 | 0.122 |
| | CV | 0.075 | 0.063 | 0.080 | 0.069 | 0.068 | 0.058 | 0.042 | 0.080 | 0.055 | 0.040 | 0.073 | 0.057 | 0.077 | 0.077 | 0.068 | 0.058 | 0.045 | 0.075 | 0.063 | 0.038 |
| | | Slice 6 | | | | | Slice 7 | | | | | Slice 8 | | | | | Slice 9 | | | | |
| | Stats | all | m/M | m/A | f/M | f/A | all | m/M | m/A | f/M | f/A | all | m/M | m/A | f/M | f/A | all | m/M | m/A | f/M | f/A |
| OD | count | 188 | 62 | 45 | 34 | 39 | 188 | 62 | 45 | 34 | 39 | 188 | 62 | 45 | 34 | 39 | 188 | 62 | 45 | 34 | 39 |
| | mean | 0.190 | 0.187 | 0.188 | 0.194 | 0.194 | 0.185 | 0.183 | 0.189 | 0.185 | 0.186 | 0.192 | 0.189 | 0.188 | 0.198 | 0.195 | 0.184 | 0.183 | 0.185 | 0.184 | 0.185 |
| | std | 0.015 | 0.014 | 0.013 | 0.014 | 0.016 | 0.010 | 0.008 | 0.012 | 0.009 | 0.012 | 0.014 | 0.013 | 0.013 | 0.014 | 0.014 | 0.010 | 0.007 | 0.011 | 0.009 | 0.012 |
| | min | 0.157 | 0.157 | 0.163 | 0.170 | 0.157 | 0.168 | 0.168 | 0.171 | 0.173 | 0.171 | 0.161 | 0.161 | 0.166 | 0.169 | 0.164 | 0.167 | 0.169 | 0.167 | 0.170 | 0.169 |
| | median | 0.190 | 0.186 | 0.190 | 0.191 | 0.195 | 0.184 | 0.181 | 0.186 | 0.184 | 0.184 | 0.191 | 0.189 | 0.188 | 0.197 | 0.192 | 0.182 | 0.182 | 0.182 | 0.182 | 0.182 |
| | max | 0.256 | 0.220 | 0.223 | 0.225 | 0.256 | 0.227 | 0.201 | 0.227 | 0.205 | 0.224 | 0.245 | 0.218 | 0.223 | 0.229 | 0.245 | 0.228 | 0.197 | 0.220 | 0.205 | 0.228 |
| | CV | 0.076 | 0.074 | 0.070 | 0.070 | 0.082 | 0.055 | 0.045 | 0.061 | 0.046 | 0.063 | 0.072 | 0.067 | 0.070 | 0.067 | 0.072 | 0.052 | 0.039 | 0.059 | 0.049 | 0.063 |
| OS | count | 186 | 65 | 39 | 38 | 36 | 186 | 65 | 39 | 38 | 36 | 186 | 65 | 39 | 38 | 36 | 186 | 65 | 39 | 38 | 36 |
| | mean | 0.190 | 0.185 | 0.189 | 0.197 | 0.193 | 0.182 | 0.180 | 0.186 | 0.183 | 0.182 | 0.191 | 0.187 | 0.188 | 0.199 | 0.195 | 0.188 | 0.186 | 0.190 | 0.191 | 0.188 |
| | std | 0.015 | 0.012 | 0.020 | 0.014 | 0.011 | 0.010 | 0.007 | 0.013 | 0.009 | 0.007 | 0.014 | 0.012 | 0.016 | 0.015 | 0.009 | 0.011 | 0.009 | 0.014 | 0.011 | 0.008 |
| | min | 0.157 | 0.163 | 0.168 | 0.172 | 0.176 | 0.158 | 0.166 | 0.166 | 0.170 | 0.169 | 0.154 | 0.160 | 0.162 | 0.173 | 0.182 | 0.167 | 0.167 | 0.171 | 0.176 | 0.172 |
| | median | 0.189 | 0.185 | 0.183 | 0.196 | 0.192 | 0.181 | 0.179 | 0.182 | 0.183 | 0.183 | 0.190 | 0.188 | 0.186 | 0.200 | 0.194 | 0.186 | 0.185 | 0.186 | 0.189 | 0.187 |
| | max | 0.268 | 0.207 | 0.268 | 0.244 | 0.219 | 0.238 | 0.198 | 0.238 | 0.205 | 0.200 | 0.236 | 0.218 | 0.233 | 0.236 | 0.215 | 0.248 | 0.203 | 0.248 | 0.233 | 0.207 |
| | CV | 0.078 | 0.062 | 0.106 | 0.069 | 0.054 | 0.053 | 0.041 | 0.069 | 0.048 | 0.039 | 0.075 | 0.065 | 0.084 | 0.076 | 0.047 | 0.057 | 0.047 | 0.072 | 0.057 | 0.041 |

OD = oculus dexter, OS = oculus sinister, Stats = statistic, std = standard deviation, min = minimum, max = maximum, CV = coefficient of variation, m = male, f = female, M = Mauritius, A = Asian, values in mm$^3$. Note that slice 1 is identical to zone 1.

Notably, the large sample size used in this study generally leads to more reliable results with greater precision and statistical power compared to studies done with a smaller number of eyes.

A limitation was that the outer demarcation of the transition between choroid and sclera was difficult to define as a result of the intense pigmentation. Therefore, it may be that the values could be slightly different. Another limitation of this study is that a 25 cross-sectional scan method included a relatively small number size to infer the three-dimensional volume of the choroid. With a higher number of cross-sectional samples, potentially, the measured volume values would be closer to the real-world values. In humans, choroid volume has been reported to increase with increasing myopia [33] which could not be assessed in this study. Thus, another possible limitation of this study is that age [33, 34], axial length, and diurnal changes were not evaluated [35, 36] and correction of these parameters was not possible. However, the assessment of these values was also not the aim of this study, and such weaknesses are typical for retrospective studies which can be improved on in future studies.

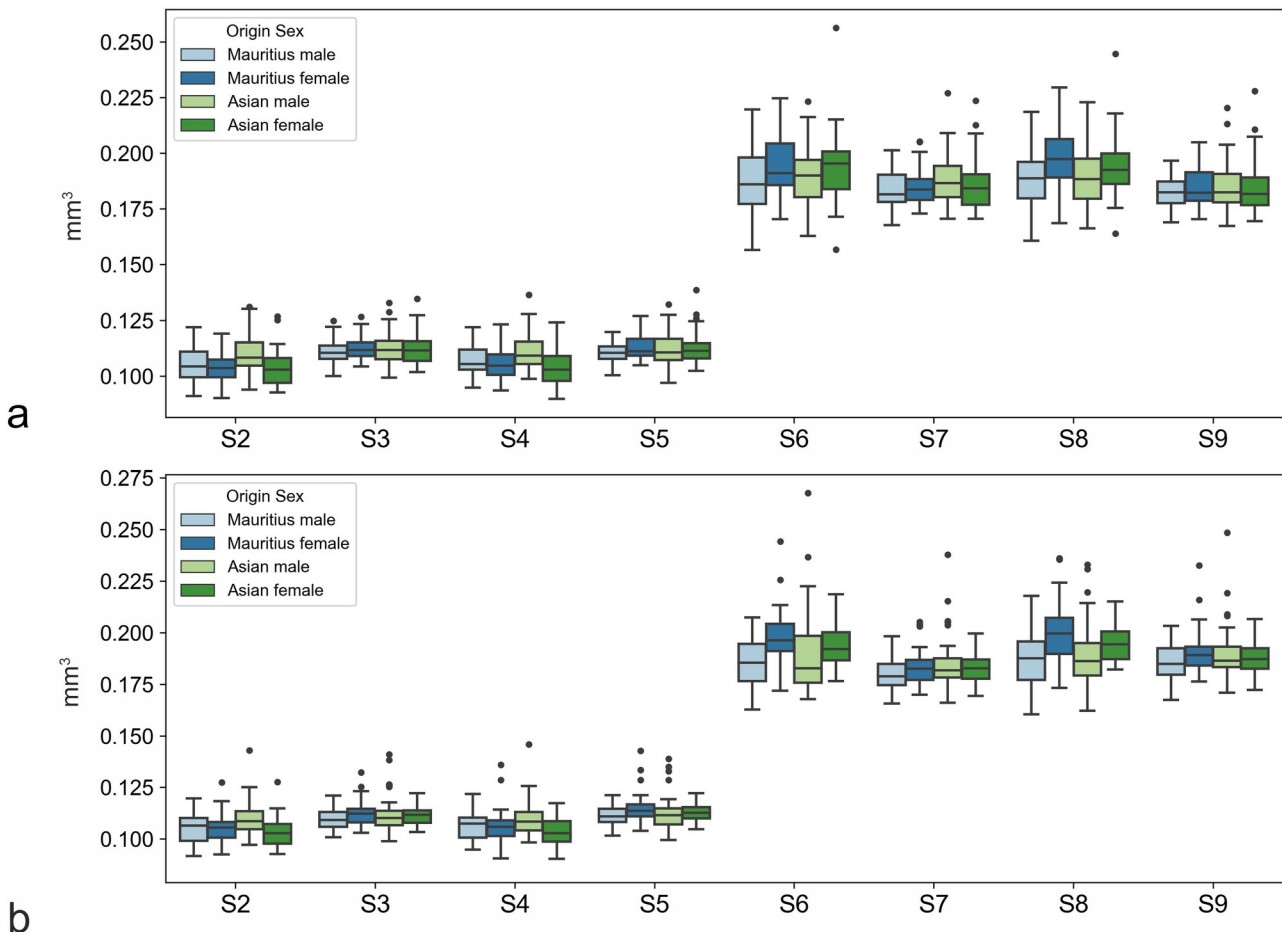

**Fig 4. Boxplots of origin-specific variations in choroid volumes.** For right (**4a**, OD) and left eyes (**4b**, OS) measured as slices.

## Conclusions

In summary, an automatic—and thus objective—hybrid ML approach showed that the choroid of cynomolgus monkeys is relatively homogonous in structure compared to the retina. The results close the gap to reduce ambiguity and difficulty in the evaluation of choroidal volume data.

**Table 4. Pearson correlation among the sixteen coefficients (3 zone, 4 quadrants, 9 slice coefficients).**

| | | | Correlation | | | |
|---|---|---|---|---|---|---|
| **Stats** | **among zones** | **among quadrants** | **among slices** | **zones and quadrants** | **zones and slices** | **quadrants and slices** |
| **mean** | 0.81 | 0.84 | 0.67 | 0.87 | 0.78 | 0.78 |
| **Min** | 0.76 | 0.82 | 0.17 | 0.80 | 0.48 | 0.52 |
| **25%** | 0.77 | 0.82 | 0.60 | 0.82 | 0.66 | 0.68 |
| **50%** | 0.80 | 0.83 | 0.69 | 0.87 | 0.85 | 0.81 |
| **75%** | 0.85 | 0.85 | 0.81 | 0.90 | 0.89 | 0.86 |
| **Max** | 0.87 | 0.88 | 0.91 | 0.93 | 1.00 | 0.98 |

Stats = statistical analysis, std = standard deviation, min = minimal, max = maximal.

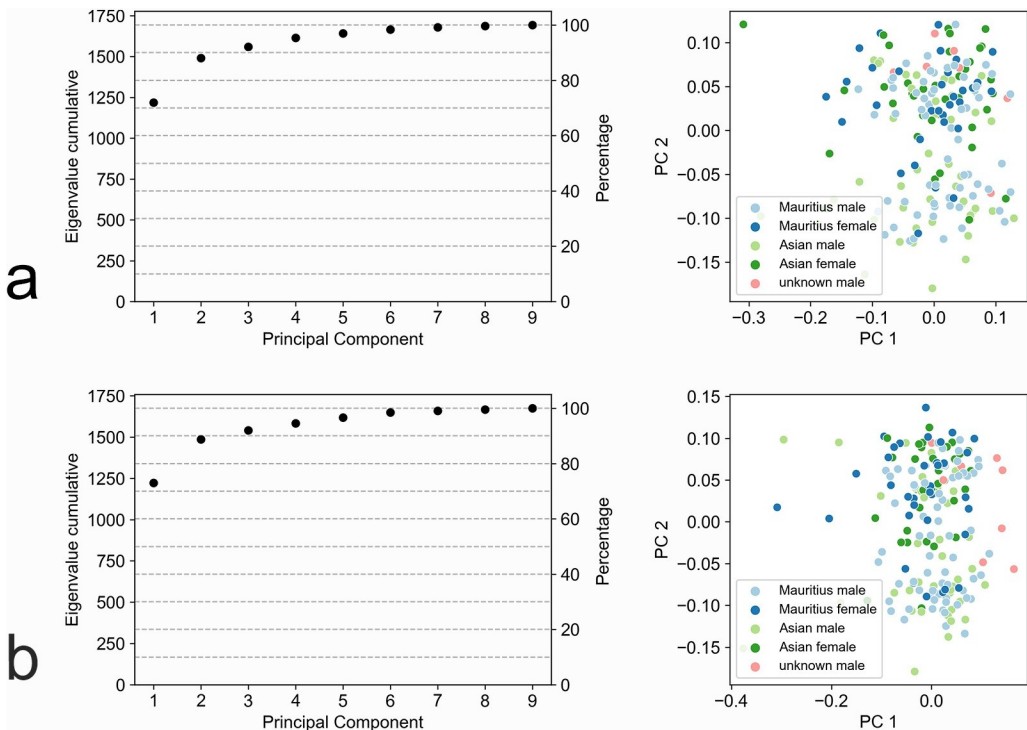

**Fig 5. Principal Component Analysis (PCA) plots of choroidal volumes S1 –S9.** (a) and (c) are scree plots showing the cumulative eigenvalues of the nine principal components (PCs) for right and left eyes, respectively. Eigenvalues indicate the explained variability of the respective PC. The first two PCs explain 88.1% and 88.8% of the variability in right and left eyes, respectively. (b) and (d) show projections of the data onto the first two principal components for right and left eyes, respectively.

**Table 5. Principal component analysis coefficients of the first two principal components for right and left eyes.**

| PC | Eye | S1 | S2 | S3 | S4 | S5 | S6 | S7 | S8 | S9 |
|---|---|---|---|---|---|---|---|---|---|---|
| 1 | Right | -12.57 | -9.88 | -12.96 | -9.77 | -13.00 | -9.78 | -12.91 | -9.79 | -13.06 |
| 2 | Right | -0.18 | -8.20 | 1.67 | -8.51 | 1.45 | 8.05 | -0.96 | 7.85 | -1.32 |
| 1 | Left | -12.70 | -9.96 | -12.98 | -10.37 | -12.74 | -10.26 | -12.77 | -9.86 | -12.55 |
| 2 | Left | -0.75 | -8.43 | 1.04 | -7.72 | 2.03 | 7.68 | -1.67 | 8.13 | -0.27 |

**Table 6. MANOVA results.**

| Eye | Variable | Wilk's lambda | *p*-value |
|---|---|---|---|
| Right | Sex | 0.7885 | p < 0.001 |
| Right | Origin | 0.8864 | 0.014 |
| Left | Sex | 0.7834 | p < 0.001 |
| Left | Origin | 0.9647 | 0.756 |

**Table 7. Summary of p-values in two-way analysis of variance (ANOVA) for measured choroidal thickness parameters in right (first two rows) and left (last two rows) eyes in relation to sex and origin.**

| | Eye | S1 | S2 | S3 | S4 | S5 | S6 | S7 | S8 | S9 |
|---|---|---|---|---|---|---|---|---|---|---|
| Sex | right | | 2.3e-3* | | 1.4e-3* | | 2.0e-3* | | 1.5e-4** | |
| Origin | right | | | | | | | | | |
| Sex | left | | 1.0e-2 | | 2.9e-3* | 8.7e-3 | 1.3e-5*** | | 1.3e-6*** | |
| Origin | left | | | | | | | | | |

*** p-values < 0.001/9.

** < 0.01/9.

* p-values < 0.05/9. Nine is the number of hypotheses and, thus, the factor applied to adjust significance levels (Bonferroni correction). Exact p-values are only shown if the results are significant.

## Supporting information

**S1 Table. The results for volume choroid data with regard to sex, origin, and eye side were included.**
(CSV)

## Acknowledgments

The authors thank Pascal Kaiser and Akiko A. Yasumoto, Zurich, Switzerland, for their support with statistical analyses.

## Author Contributions

**Conceptualization:** Peter M. Maloca, Marlene Juedes, Pascal W. Hasler, Hendrik P. N. Scholl, Nora Denk.

**Data curation:** Peter M. Maloca, Philippe Valmaggia, Theresa Hartmann, Marlene Juedes, Nora Denk.

**Formal analysis:** Peter M. Maloca, Philippe Valmaggia, Pascal W. Hasler, Hendrik P. N. Scholl, Nora Denk.

**Funding acquisition:** Peter M. Maloca, Hendrik P. N. Scholl, Nora Denk.

**Investigation:** Peter M. Maloca, Theresa Hartmann, Marlene Juedes, Nora Denk.

**Methodology:** Peter M. Maloca, Philippe Valmaggia, Theresa Hartmann, Pascal W. Hasler, Hendrik P. N. Scholl, Nora Denk.

**Project administration:** Peter M. Maloca, Theresa Hartmann, Nora Denk.

**Resources:** Peter M. Maloca, Marlene Juedes, Hendrik P. N. Scholl, Nora Denk.

**Software:** Peter M. Maloca.

**Supervision:** Peter M. Maloca.

**Validation:** Peter M. Maloca, Nora Denk.

**Visualization:** Peter M. Maloca.

**Writing – original draft:** Peter M. Maloca, Philippe Valmaggia, Theresa Hartmann, Marlene Juedes, Pascal W. Hasler, Hendrik P. N. Scholl, Nora Denk.

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
