## [Decision Letter · Decision Letter 0]

28 Jul 2022

PONE-D-22-14925Volumetric subfield analysis of cynomolgus monkey’s choroid derived from hybrid machine learning optical coherence tomography segmentationPLOS ONE

Dear Dr. Maloca,

Thank you for submitting your manuscript to PLOS ONE. After careful consideration, we feel that it has merit but does not fully meet PLOS ONE’s publication criteria as it currently stands. Therefore, we invite you to submit a revised version of the manuscript that addresses the points raised during the review process.

Both reviewers felt that your research was technically sound and that your results supported your conclusions. Please provide more details concerning your statistical methods and make the text changes recommended by the two expert reviewers.

We look forward to receiving your revised manuscript.

Kind regards,

Alfred S Lewin, Ph.D.

Section Editor

PLOS ONE

Journal Requirements:

Reviewers' comments:

Reviewer's Responses to Questions

**Comments to the Author**

1. Is the manuscript technically sound, and do the data support the conclusions?

Reviewer #1: Yes

Reviewer #2: Yes

2. Has the statistical analysis been performed appropriately and rigorously? 

Reviewer #1: Yes

Reviewer #2: Yes

3. Have the authors made all data underlying the findings in their manuscript fully available?

Reviewer #1: Yes

Reviewer #2: Yes

4. Is the manuscript presented in an intelligible fashion and written in standard English?

Reviewer #1: Yes

Reviewer #2: Yes

5. Review Comments to the Author

Reviewer #1: The article studied the influence of sex, origin and eye sites on the choroid volume of cynomolgus monkeys. Some minor issues should be addressed.

1. The “Materials and methods” section was not clearly elaborated. The software and detailed version used for Pearson correlation analysis and Principal Component Analysis (PCA) should be addressed in the 2nd paragraph in “Statistical analysis”.

2. The definition and value of significance level of MANOVA and ANOVA tests should be stated in the “Materials and methods” section.

3. In the “Results” section, the paragraph of “MANOVA analysis”, note the grammar mistake of the sentence: “A joint analysis of the nine slice coefficients (S1 - S9) appears reasonable because they nine slice coefficients are correlated with each other (see correlation analysis).”

4. Suggest, putting figure legends in the upper left corner of Figure 3, same as other figures.

5. Please add scatterplots against the first two PCs for right and left eyes to show the clustering of choroidal volumes S1 – S9 in Figure 5.

6. The biological significance and implications of the study should be mentioned in the “Discussion” section.

Reviewer #2: In this manuscript ‘Volumetric subfield analysis of cynomolgus monkey’s choroid derived from hybrid machine learning optical coherence tomography segmentation’, the authors described choroidal volume in monkey eyes which was measured using OCT and ML algorithm.

This study has its importance in providing standards for choroidal volume metrics in monkey eyes since it includes relatively large number of cynomolgus monkeys.

Abstract

Methods should describe the experiment included both Asian or Mauritius) cynomolgus macaques.

Introduction

Line 94, the foveas -> the fovea

Line98-99 photoreceptor cells are supplied by choriocapillaris. Absence of retinal vessel does not mean they work in hypoxic zone.

Methods

Line166 25 rater line scan using Spectral OCT system is not a good option to study choroidal volume.

ML algorithms used in choroidal volume measurement are poorly described. And rationale for using ML algorithm rather than manual measurement should be described.

Results

Line252, ‘the region of the sharpest vision’, could be removed.

Line256 Please provide p-values for tables 1 and 2 if there were any significant differences between groups.

Table6, what causes differences in effect of origin in right and left eye?

The monkeys d

Discussion

The first paragraph of the discussion has numerous sentences which are duplicated from the introduction.

Is there any explanation for differences in choroidal volume between sex?

6. PLOS authors have the option to publish the peer review history of their article (what does this mean?). If published, this will include your full peer review and any attached files.

Reviewer #1: No

Reviewer #2: No

---

## [Author Response · Author response to Decision Letter 0]

31 Aug 2022

Point-to-point response

“Volumetric subfield analysis of cynomolgus monkey’s choroid derived from hybrid machine learning optical coherence tomography segmentation”

Comments to reviewer #1

The article studied the influence of sex, origin and eye sites on the choroid volume of cynomolgus monkeys. Some minor issues should be addressed.

Comment 1: The “Materials and methods” section was not clearly elaborated. The software and detailed version used for Pearson correlation analysis and Principal Component Analysis (PCA) should be addressed in the 2nd paragraph in “Statistical analysis”.

Response 1: We added the detailed software version used for the Principal Component Analysis (PCA) and the Pearson correlation analysis. Thank you for pointing out.

Comment 2: The definition and value of significance level of MANOVA and ANOVA tests should be stated in the “Materials and methods” section.

Response 2: We added a description of the significance levels used in MANOVA and ANOVA tests to the “Materials and methods” section.

Comment 3: In the “Results” section, the paragraph of “MANOVA analysis”, note the grammar mistake of the sentence: “A joint analysis of the nine slice coefficients (S1 – S9) appears reasonable because they nine slice coefficients are correlated with each other (see correlation analysis).”

Response 3: Thank you very much for pointing out. We corrected the mistake.

Comment 4: Suggest, putting figure legends in the upper left corner of Figure 3, same as other figures.

Response 4: We did as the reviewer suggested. Thank you for pointing out. Having the figure legends in the same place for Figures 2, 3, and 4 is a very good idea.

Comment 5: Please add scatterplots against the first two PCs for right and left eyes to show the clustering of choroidal volumes S1 – S9 in Figure 5.

Response 5: We added the scatterplots showing a projection of the data onto the first two principal components as requested. However, no obvious clusters with respect to sex and/or origin are apparent from the scatter plots (this was also the reason why we did not include these plots in the first version of the manuscript draft). However, adding the additional plots is perfectly fine. A principal component analysis does not always yield principal components that are informative with respect to categorical variables that divide individual data point into categories.

Comment 6: The biological significance and implications of the study should be mentioned in the “Discussion” section.

Response 6: The monkeys studied are not free-living animals. These animals were bred for laboratory studies and come from two different geographic regions (Mauritius and Asia Origin as listed in the manuscript). Hence, they are not free-living animals. Therefore, we think that biologically there is only a marginal significance. However, laboratory-technically we have found significant differences that were not known before. Thus, our study is very relevant for the design and conduct of pre-clinical studies to prevent or be aware of a selection bias, so that no misinterpretation is made from the results.

Comments to reviewer #2

In this manuscript ‘Volumetric subfield analysis of cynomolgus monkey’s choroid derived from hybrid machine learning optical coherence tomography segmentation’, the authors described choroidal volume in monkey eyes which was measured using OCT and ML algorithm. This study has its importance in providing standards for choroidal volume metrics in monkey eyes since it includes relatively large number of cynomolgus monkeys.

Comment 7: Methods should describe the experiment included both Asian or Mauritius) cynomolgus macaques.

Response 7: We added a corresponding additional clause to the abstract stating that our experiments used macaques of Asian and Mauritian origin.

Comment 8: Introduction Line 94, the foveas -> the fovea

Response 8: We corrected the mistake. Thank you for pointing out.

Comment 9: Line98-99 photoreceptor cells are supplied by choriocapillaris. Absence of retinal vessel does not mean they work in hypoxic zone.

Response 9: This is absolutely true. Thank you for pointing this out. We have adjusted the manuscript: 

“This is particularly remarkable, because the photoreceptor cells are dependent on a healthy choriocapillaris.” 

Comment 10: Methods, Line 166, 25 rater line scan using Spectral OCT system is not a good option to study choroidal volume.

Response 10: We agree with the reviewer that having more than 25 raster line scans per OCT scan would be advantageous to study choroid volumes. In preclinical 

studies using macaque models systems, 25 raster line scans appear to be a standard. In this retrospective study, we do not have the possibility to change this setting. However, we try to emphasize this limitation more in the Discussion. We adjusted the wording to:

“Another limitation of this study is that a 25 cross-sectional scan method included a relatively small number size to infer the three-dimensional volume of the choroid. With a higher number of cross-sectional samples, potentially, the measured volume values would be closer to the real-world values.”

Comment 11: ML algorithms used in choroidal volume measurement are poorly described. And rationale for using ML algorithm rather than manual measurement should be described.

Response 11: We improved the description of the machine learning algorithm. In addition, we provide some rational for using a machine learning algorithm rather than manual annotations by human annotators:

“In short, the machine learning algorithm is a scalable, deep learning-based algorithm which creates semantic image segmentations of B-scans [25]. For every pixel of a B-scan it predicts the eye compartment, i.e., vitreous, retina, choroid, or sclera (Fig 1a). This study used the deep learning-based algorithm to identify the choroid compartment. The deep learning-based algorithm generated a semantic image segmentation of a B-scan within seconds, whereas a human grader generally needs longer and gets tired after some time.”

Comment 12: Results, Line 252, ‘the region of the sharpest vision’, could be removed.

Response 12: Thank you for pointing out. We think, however, that it could be beneficial if the clause remained in the manuscript. In the corresponding paragraph, we are reporting summary statistics of zone 1. We do not report summary statistics for other zones because, in some sense, the other zones are not as important as zone 1. Zone 1 is the region of sharpest vision which tends to get more attention in ophthalmologic examinations. We thus think it could be helpful to keep the clause to highlight to the reader that the summary statistics relate to zone 1, the region of sharpest vision.

Comment 13: Line 256 Please provide p-values for tables 1 and 2 if there were any significant differences between groups.

Response 13: Thank you for this advice. Providing p-values for Table 1 and 2 seems like a reasonable thing to do at first. Table 1, 2, and 3 are similar to each other since they contain the same type of data for zones, quadrants, and slices, respectively. For the slices in Table 3 we provide p-values (Table 7), but we do not provide p-values for the zones and quadrants in Table 1 and 2. Why do we do that? The reason is (1) the high correlation among zones/quadrants and slices and (2) the multiple testing problem. The more statistical hypothesis tests we perform the higher is the probability to get a false-positive test result (this is also why we do the Bonferroni correction). In addition, the zones/quadrants (Table 1 and 2) are so highly correlated to the slices (Table 4) that the additional statistical hypothesis tests would be highly non-independent from the tests on the slices. We therefore decided not to include statistical hypothesis tests on the zones and quadrants in this manuscript. 

The same kind of reasoning, in somewhat different words, is also put forward in the manuscript in the Results section under Correlation Analysis.

Comment 14: Table 6, what causes differences in effect of origin in right and left eye?

Response 14: We believe the difference in p-values (0.014 vs 0.756) is caused by chance. The p-values basically tell us the probability that the data we observe (or data that is “more extreme” than ours) could have occurred by chance. A p-value of 0.014 is admittedly quite small. However, it is above the significance level of 0.01. In recent years, in applied statistics, there was a general trend to move from significance level 0.05 to 0.01 to reduce the chance of false positives. With significance level 0.05, one out of 20 studies would report false findings. With 0.01, this is reduced to one out of 100. In the context of the current replication crisis in science, using significance level 0.01 is probably a reasonable thing to do.

Comment 15: Discussion: The first paragraph of the discussion has numerous sentences which are duplicated from the introduction.

Response 15: We understand this comment very well. The reality is probably that most readers jump from the abstract directly to the discussion. Thus, we wanted to provide a brief guide for the reader to help them read the article so they don't have to start out of nowhere. Nevertheless, thanks to this good comment, we have shortened the discussion to reduce duplicates. We hope that the reviewer is satisfied with the new and more readable version

Comment 16: Is there any explanation for differences in choroidal volume between sex?

Response 16: Our data show that sex only just had a significant effect on choroidal volumes in the superior-inferior axis, but not in the fovea centralis. We have no explanation for this and can only speculate. In any case, the supply of the fovea from the choroid is without major differences between the sexes.

---

## [Editor Report · Decision Letter 1]

12 Sep 2022

Volumetric subfield analysis of cynomolgus monkey’s choroid derived from hybrid machine learning optical coherence tomography segmentation

PONE-D-22-14925R1

Dear Dr. Maloca,

We’re pleased to inform you that your manuscript has been judged scientifically suitable for publication and will be formally accepted for publication once it meets all outstanding technical requirements. Thank you for carefully responding to the issues raised by the reviewers.

Kind regards,

Alfred S Lewin, Ph.D.

Section Editor

PLOS ONE
---

## [Editor Report · Acceptance letter]

14 Sep 2022

PONE-D-22-14925R1 

Volumetric subfield analysis of cynomolgus monkey’s choroid derived from hybrid machine learning optical coherence tomography segmentation 

Dear Dr. Maloca:

I'm pleased to inform you that your manuscript has been deemed suitable for publication in PLOS ONE. Congratulations! Your manuscript is now with our production department. 

Kind regards, 

on behalf of

Dr. Alfred S Lewin 

Section Editor

PLOS ONE